

# Laboratory rivers:
# Lacey's law, threshold theory and channel stability

François Métivier[1], Eric Lajeunesse[1], and Olivier Devauchelle[1]

[1]Institut de physique du globe de Paris – Sorbonne Paris Cité, Université Paris Diderot, CNRS, UMR7154, 1 rue Jussieu, 75238 Paris Cedex 05, France

*Correspondence to:* F. Métivier (metivier@ipgp.fr)

## Abstract

More than a century of experiments have demonstrated that many features of natural rivers can be reproduced in the laboratory. Here, we revisit some of these experiments to cast their results into the framework of the threshold-channel theory developed by Glover and Florey (1951). In all the experiments we analyze, the typical size of the channel conforms to this theory, regardless

of the river's planform (single-thread or braiding). In that respect, laboratory rivers behave exactly like their natural counterpart. Using this finding to reinterpret experiments by Stebbings (1963), we suggest that sediment transport widens the channel until it reaches a limit width, beyond which it destabilizes into a braided river. If confirmed, this observation would explain the remarkable scarcity of single-thread channels in laboratory experiments.

## 1   Introduction

At the turn of the 20th century, Jaggar (1908) developed a series of laboratory experiments to produce small-scale analogues of rivers (figure 1a). In the first one, a subsurface flow seeps out of a layer of sediment. Sapping then erodes the sediment, and this process generates wandering channels. Introducing rainfall in another experiment, he was able to generate a ramified network of small rivers, which drains water out of the sediment layer, much like a natural hydrographic network drains rainwater out of its catchment. The similarity between his experiments and natural systems led Jaggar to the following conclusion (Jaggar,

1908, p. 300):

> *The foregoing experiments suggest many questions and answer few. They are based on the assumption that the extraordinary similarity of the rill pattern to the mapped pattern of rivers is due to government in both cases by similar laws.*

Jaggar was therefore convinced that we should use laboratory analogues to investigate, under well-controlled conditions, the

mechanisms by which a river forms, and how it selects its geometry.

Forty years later, Friedkin (1945) used a laboratory flume to investigate the stability of a river's course. In his experiment, he carved a straight channel in a layer of sand, and sharply curved its course near the water inlet. This perturbation causes the



channel to erode its banks and migrate laterally. As it does so, the channel becomes sinuous, and a well-defined wavelength emerges (figure 1b). Friedkin then explored systematically the influence of the control parameters (grain size, initial geometry, water and sediment discharge) on this response. His observations showed that water and sediment discharges are the main control on the channel's cross section and planform geometry. In particular, when the sediment discharge gets large, the channel

turns into a braided river. Conversely, in the absence of sediment load, the channel relaxes towards an isolated steady thread.

Building on Friedkin's work, Leopold and Wolman (1957) located, in the parameter space, the braiding transition of a laboratory channel. To do so, they supplied water and sand to an initially straight channel. As this channel adapts to the input, mid-channel bars form which tend to separate the flow, and eventually split the channel. Ultimately, the experiment generates a braided river. Leopold and Wolman then observed that braided threads have, on average, a larger longitudinal slope than their

isolated counterparts. Inspired by this finding, they plotted field observations on a slope-discharge diagram, and showed that braided channels are separated from single-thread ones by a critical value of the slope $S_c$, which decreases with discharge $Q$ according to $S_c = 0.06 Q^{-0.44}$ (discharge in ft$^3$s$^{-1}$).

To our knowledge, such an empirical boundary has never been drawn for laboratory experiments, partly because maintaining an active single-thread channel has proven an experimental challenge (Schumm et al., 1987; Murray and Paola, 1994; Federici

and Paola, 2003; Paola et al., 2009). In non-cohesive sediment, most experimental channels turn into a braided river, unless they do not transport any sediment. This propensity for braiding persists when the water discharge varies during the experiment, and seems unaffected by grain size (Sapozhnikov and Foufoula-Georgiou, 1996, 1997; Métivier and Meunier, 2003; Leduc, 2013; Reitz et al., 2014).

By contrast, preventing bank erosion helps maintaining a single-thread channel. One way to do so is to add some fine and

cohesive sediment to the mixture injected into the experiment (Schumm et al., 1987; Smith, 1998; Peakall et al., 2007; Dijk et al., 2012). Another successful method is to grow riparian vegetation on the emerged areas of the flume. Tal and Paola (2007) and Brauderick et al. (2009) used alfalfa sprouts, which roots protect the sediment they grow upon from scouring. These observations show that bank cohesion, in addition to sediment discharge, controls the planform geometry of laboratory rivers. However, the relative importance of these parameters remains debatable, both for laboratory experiments and for natural rivers

(Métivier and Barrier, 2012). To address this question, we need to formalize, in a suitable theoretical framework, the interplay between the dynamics of sediment transport and the mechanical stability of a channel's banks.

To design stable irrigation canals, Glover and Florey (1951) calculated the shape of a channel which bed is at the threshold of motion. Henderson (1963) referred to this work as the threshold theory, and showed that it applies to natural rivers as well. This theory offers a physical interpretation for the empirical relationship proposed by Lacey (1930), according to which the

width of an alluvial river increases in proportion to the square root of its water discharge (Henderson, 1963; Andrews, 1984; Devauchelle et al., 2011b; Gaurav et al., 2015; Métivier et al., 2016).

In a series of theoretical papers, Parker and coauthors extended the threshold theory to active alluvial rivers, that either maintain their banks at the threshold of sediment motion, or rebuild them constantly by depositing a fraction of their suspended load (Parker, 1978a, b, 1979; Kovacs and Parker, 1994). These mechanisms counteract the bank collapse induced by gravity,

and the resulting balance controls the geometry of their bed. This theory provides a physical basis for comprehensive regime





relations, which describe the geometry of alluvial rivers as a function of their water and sediment discharges (Parker et al., 2007). Does this theoretical framework equally apply to laboratory rivers?

Here, we investigate this question by re-interpreting experiments performed since the late 1960s in the light of the threshold theory. We begin with a brief presentation of the connection between Lacey's law and this theory, and then evaluate its appli-
cability to laboratory experiments (section 2). Finally, using the experimental observations of Stebbings (1963), we propose an empirical criterion for the stability of an active channel in non-cohesive sediment, and compare it to laboratory single-thread and braided channels (Ikeda et al., 1988; Ashmore, 2013) (section 3).

## 2   Lacey's law and the threshold theory

In 1930, Lacey remarked that irrigation canals remain stable when their width scales as the square root of their discharge, even
when they are cut into loose material (Lacey, 1930). Field observation later revealed that Lacey's law applies to natural rivers as well. For illustration, we use the compendium of Li et al. (2015) to plot the width of a broad range of alluvial rivers against their water discharge (Figure 2a). Over twelve orders of magnitude in discharge, the data points gather around a 1/2 power law, in accordance with Lacey's law.

Lacey's relationship remained an empirical law until Glover and Florey (1951) calculated the cross-section shape of a
channel which bed is at the threshold of motion. When the water flow is just strong enough to entrain the bed material, the balance between gravity and fluid friction sets the cross-section shape and the downstream slope of the channel. In particular, this balance relates the width $W$ of a channel to its discharge $Q$ (Glover and Florey, 1951; Henderson, 1963; Devauchelle et al., 2011b; Seizilles, 2013):

$$\frac{W}{d_s} = \left[ \frac{\pi}{\sqrt{\mu}} \left( \frac{\theta_t(\rho_s - \rho)}{\rho} \right)^{-1/4} \sqrt{\frac{3C_f}{2^{3/2}\mathcal{K}\left[1/2\right]}} \right] Q_*^{1/2}, \tag{1}$$

where $Q_* = Q/\sqrt{gd_s^5}$ is the dimensionless discharge, $d_s$ is the grain size of the sediment, $\rho$ and $\rho_s$ are the densities of water and of the sediment, $C_f$ is the turbulent friction coefficient, $\theta_t$ is the threshold Shields parameter, $\mu$ is the friction angle, and finally $\mathcal{K}[1/2] \approx 1.85$ is a transcendental integral.

Glover's and Florey's theory explains the exponent of Lacey's law, but what about its prefactor? Some of the parameters in the prefactor of eq. (1) are approximately constant in nature: the density of water ($\rho \simeq 1000\,\mathrm{kg/m^3}$), that of sediment
($\rho_s \simeq 2650\,\mathrm{kg/m^3}$), and the friction angle ($\mu \simeq 0.7$). Other ones vary significantly. For instance, the median grain size $d_{50}$ extends over three orders of magnitudes in the data set we use ($0.1\,\mathrm{mm} - 10\,\mathrm{cm}$). In addition, the sediment is often broadly distributed in size within a river reach which, strictly speaking, impairs the applicability of the threshold theory. We do not know how a broad grain-size distribution affects eq. (1). For lack of a better solution, hereafter we use the median of the distribution as an approximation of the grain size ($d_s \simeq d_{50}$). Similarly, the value of the turbulent friction coefficient $C_f$ typically extends
over almost two orders of magnitude in nature ($0.02 - 0.1$), depending on the flow Reynolds number and the bed roughness (Buffington and Montgomery, 1997). The Shields parameter $\theta_t$ varies between about $0.03$ and $0.3$, depending on the Reynolds number at the grain's scale (Recking et al., 2008; Andreotti et al., 2012; Li et al., 2015). One can take these variations into



**Figure 1.** Examples of laboratory rivers. White arrows denote flow direction. Scales are approximate. **(a)** Sapping channels (adapted from plate 1p, Jaggar, 1908). **(b)** Sinuous channel in sandy bed (adapted from plate 3, Friedkin, 1945). **(c)** Meandering channel forced by the oscillation of the inlet (Dijk et al., 2012). **(d)** Metamorphosis of a braided river into a single-thread channel induced by vegetation (Tal and Paola, 2010, with permission from John Wiley & Sons). **(e)** Active braided river in coarse sand (Leduc, 2013).





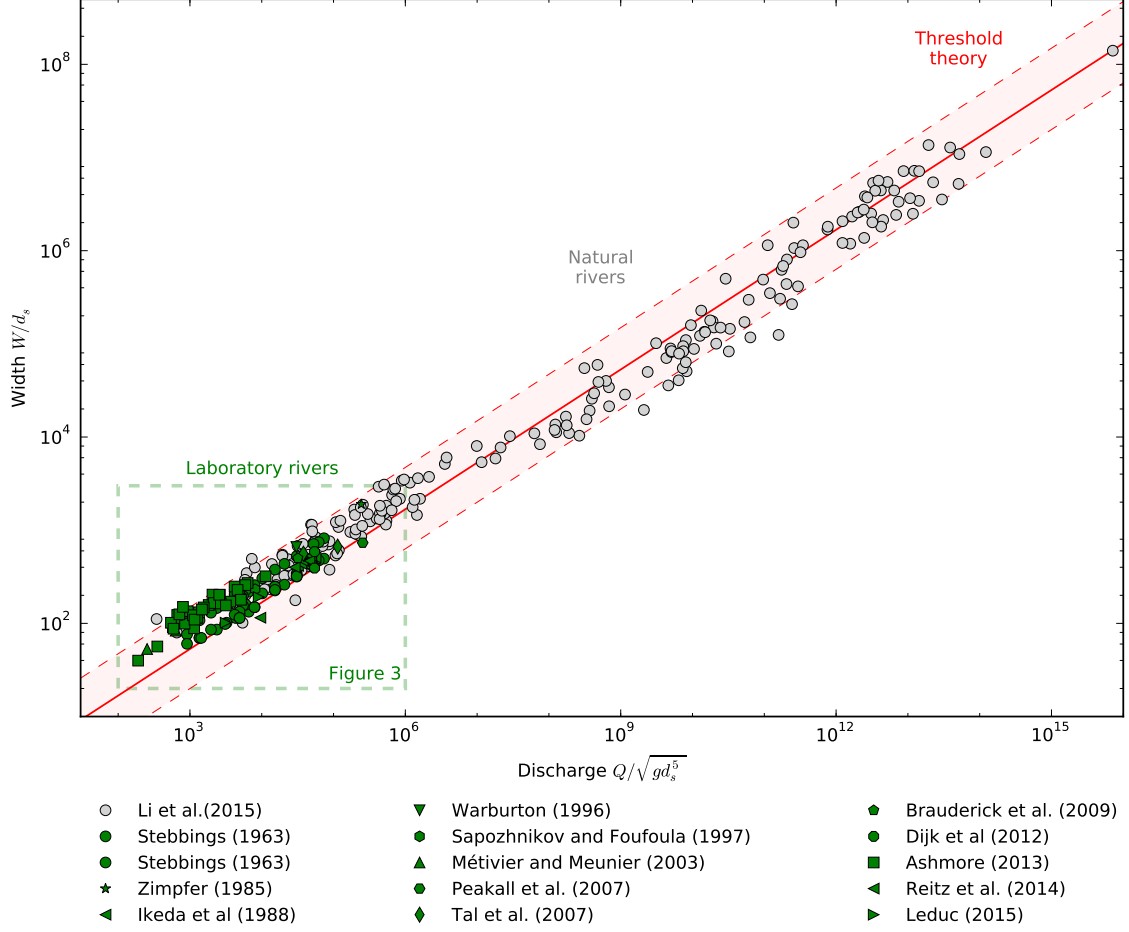

**Figure 2.** Lacey's law compared to the threshold theory, for natural rivers (gray) and laboratory rivers (green). Dimensionless width $W/d_s$ as a function of dimensionless discharge $Q/\sqrt{gd_s^5}$. Red line corresponds to the threshold theory (eq. (1) with $\theta_t = 0.05$ and $C_f = 0.1$). Shaded area and dashed lines indicate uncertainty about the parameters.

account by supplementing eq. (1) with empirical expressions that relate $C_f$ and $\theta_t$ to the water depth and median grain size (Parker et al., 2007). However, the rough approximation we use for the grain size would make such exactitude superfluous. Accordingly, we simply evaluate eq. (1) using typical values for its parameters ($\rho = 1000\,\mathrm{kg/m^3}$, $\rho_s = 2650,\mathrm{kg/m^3}$, $\theta_t = 0.05$, $C_f = 0.1$), and represent the impact of their variability as an uncertainty on the prediction (Figure 2).

5    Virtually all rivers from the compendium of Li et al. (2015) fall within this uncertainty. Equation (1) provides a reasonable first-order estimate of the size of a river, thus supporting Henderson's hypothesis: the force balance at the grain's scale explains Lacey's relationship (Henderson, 1963; Andrews, 1984; Savenije, 2003; Devauchelle et al., 2011a). Recent experiments involv-





ing a laminar flume have shown it possible to reproduce this balance in the laboratory (Seizilles et al., 2013). More generally, though, do laboratory rivers conform to the threshold theory, like their natural counterpart?

To answer this question, we compiled data from a variety of laboratory experiments (table 1, figure 1). We selected a broad range of experimental conditions, and included as many shapes of channel as possible (braided, straight, sinuous). Of

course, our choice was limited to contributions that fully report experimental conditions and observations, either explicitly or in the form of figures. Among these experiments, many generated braided rivers. We treated the individual threads of these as independent channels, as proved instructive for the interpretation of field data (Gaurav et al., 2015; Métivier et al., 2016). We find that the width of all the laboratory channels we selected conforms well to Lacey's law (figure 2). In fact, the laboratory experiments partly overlap the compendium of Li et al. (2015) and, where they do, experimental channels cannot be

distinguished from natural rivers. In that sense, laboratory rivers do not just resemble natural ones, but rather are small rivers on their own right.

Figure 2 shows that laboratory rivers select their own size according to the available water discharge, like natural rivers do. As a consequence, the threshold theory provides a reasonable estimate of their size, regardless of the specifics of each experiment. This robustness is again reminiscent of Lacey's law, which holds under a variety of natural conditions.

All this, of course, is excellent news for experimental geomorphology. If indeed experimental flumes are but small rivers, the understanding we gain in the laboratory is likely to apply in nature. This continuity, however, revives an old question: How can single-thread channels be so difficult to maintain experimentally, whereas they are ubiquitous in nature? In the next section, we investigate the stability of a single-thread channel by revisiting the laboratory observations of Stebbings (1963).

## 3 Channel stability

The elusiveness of the single-thread channel lead some authors to the conclusion that laboratory experiments lack a vital ingredient, such as sediment cohesion or vegetation, to generate realistic rivers (Schumm et al., 1987; Smith, 1998; Peakall et al., 2007; Dijk et al., 2012; Tal and Paola, 2007; Brauderick et al., 2009). This view parallels a more conceptual criticism of the threshold theory: by definition, it cannot take sediment transport into account. Indeed, an arbitrarily small amount of mobile sediment can, in principle, destabilize the threshold channel (Parker, 1978b). What specific mechanism maintains the

bed of single-thread rivers in nature remains a matter of debate. In this section, we propose a detailed comparison of laboratory channels with the threshold theory, hoping it will help us address this question.

We now return to the diagram of figure 2, and focus on laboratory experiments (figure 3). This closer view reveals that laboratory channels follow two distinct trends, depending on their planform geometry. The data points corresponding to single-thread channels align with the threshold theory (the parameters in equation (1) correspond to the experiment of Stebbings

(1963)). Conversely, the threads of braided rivers tend to be wider than predicted, although they also follow a square-root relationship. These two distinct trends emerge from a large collection of disparate experiments. We thus interpret them as the signature of an underlying common parameter that determines the planform geometry of a channel, and affects the prefactor of Lacey's law.




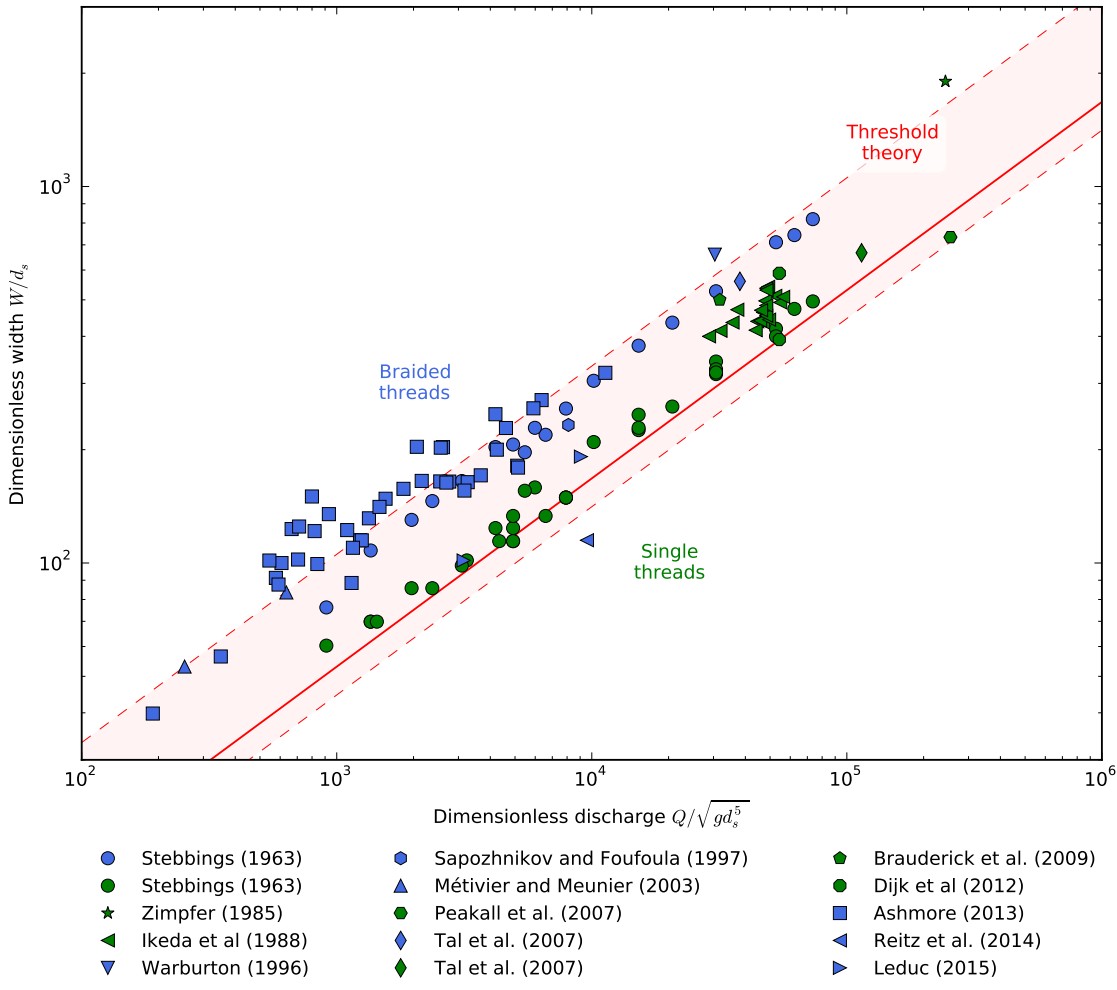

**Figure 3.** Lacey's law and threshold theory in laboratory experiments. Green: single-thread channels; blue: threads from braided rivers. Red line corresponds to the threshold theory (eq. (1) with $\theta_t = 0.05$ and $C_f = 0.1$). Shaded area and dashed lines indicate uncertainty about the parameters in experiments.



| Source | River type | Flume size $L \times W$, [m] | Grain size $d_s$, [mm] | Re | Fr | Re$_p$ |
|---|---|---|---|---|---|---|
| Stebbings (1963) | Threshold & Braided | $8 \times 0.9$ | 0.8 | 3000 | 0.7 | 130 |
| Zimpfer (1985) in Schumm et al. (1987) | Straight | $30 \times 7$ | 0.56 | 5300 | 0.92 | 41 |
| Ikeda et al. (1988) | Straight | $15 \times 0.5$ | 1.3 | 14200 | 0.5 | 150 |
| Warburton (1996) | Braided | $20 \times 3$ | 0.5 | 2000-3000 | 0.68-0.9 | 30-40 |
| Sapozhnikov and Foufoula-Georgiou (1996) | Braided | $5 \times 0.75$ | 0.12 | - | - | 4 |
| Métivier and Meunier (2003) | Braided | $1 \times 0.5$ | 0.5 | 150 | 2 | 30 |
| Tal and Paola (2007) | Sinuous & Braided | $16 \times 2$ | 0.5 | 2000-9000 | 1 | 35 |
| Peakall et al. (2007) | Sinuous | $5.5 \times 3.7$ | 0.21 | 4500 | 0.79 | 36 |
| Brauderick et al. (2009) | Sinuous | $17 \times 6.7$ | 0.8 | 4500 | 0.55 | 70 |
| Dijk et al. (2012) | Sinuous | $11 \times 6$ | 0.51 | 3300 | 0.58 | 33 |
| Ashmore (2013) | Braided | $10 \times 2$ | 1.67 | 1000-4000 | 1 | 220 |
| Leduc (2013) | Braided | $5 \times 1$ | 1.3 | 600-1000 | 0.8-1 | 24-45 |
| Reitz et al. (2014) | Braided | $1.5 \times 0.75$ | 0.26 | 250 | 2 | 15 |

**Table 1.** Experimental setups and flow conditions for the studies used in the present article. Flow conditions are characterized by the Reynolds number $\mathrm{Re} = VH/\nu$, the Froude number $\mathrm{Fr} = \sqrt{V^2/gH}$ and the particle Reynolds number $\mathrm{Re_p} = \sqrt{gd_s}d_s/\nu$ ($V$ is the mean flow velocity, $H$ is the mean channel depth, $d_s$ is the mean grain size, $g$ is the acceleration of gravity, and $\nu$ is the kinematic viscosity).

To isolate this prefactor in the laboratory, the ideal experiment would produce single-thread and braided rivers under similar conditions. The flume experiment of Stebbings (1963) approaches this ideal. Stebbings simply carved a straight channel in a flat bed of well-sorted sand. He then let a constant flow of water run into this channel, which morphology gradually adjusted to the water discharge (Figure 4). Before reaching steady state, however, the river undergoes a reproducible transient. The flow first incises the channel near the inlet, and entrains the detached sediment towards the outlet. As a result, bedload transport intensifies downstream. Stebbings noted that the river responds to this increase by widening its channel. In some cases, a bar emerges near the center of the widened channel, and the river turns into a braid. If, following Stebbings, we assume that the channel cross-section adjusts to the local sediment discharge, then his transient channel materializes the transition of a river from a channel at threshold to a collection of braided threads. Although unconfirmed yet, the hypothesis that the sediment load triggers the metamorphosis of a river has been proposed previously to interpret field observations (Mackin, 1948; Smith and Smith, 1984; Métivier and Barrier, 2012).

Once the channel has reached steady state, it does not transport anymore sediment, and we can expect it to be exactly at threshold. We indeed find that the size of Stebbings' steady-state channels accords well with the threshold theory (figure 3). This also holds, albeit less literally so, for their depth and downstream slope (Appendix A). A better way to evaluate this agreement is to correct the width from the infulence of discharge. To do so, we introduce the detrended width $W_*$ as the ratio





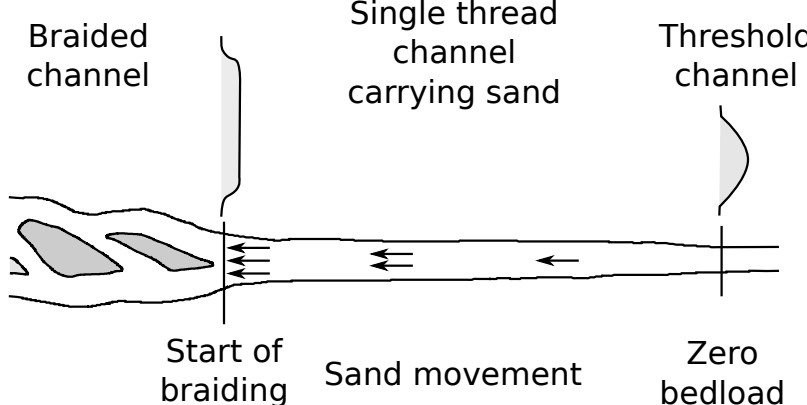

**Figure 4.** Transient channel in Stebbings's experiment (reproduced from Stebbings (1963)). Flow from right to left.

of the channel width to the width predicted by the threshold theory (Gaurav et al., 2015):

$$W_* = \frac{W}{C_W d_s \sqrt{Q_*}} \tag{2}$$

where $C_W$ is the prefactor between brackets in eq. (1). For a threshold channel, we expect $W_*$ to be one regardless of water discharge. Unsurprisingly, $W_*$ shows no dependency on discharge for the steady-state channels of Stebbings (1963) (figure 5).

5    Its average is $\langle W_* \rangle = 1.07 \pm 0.16$, confirming the accord of Stebbings' measurements with the threshold theory.

     We now turn our attention to active channels (i.e. channels transporting sediment). In Stebbings' experiment, the channel is active during the transient, and we expect its width to deviate from that of the threshold channel. The downstream widening of the river indicates that sediment transport tends to induce a wider channel (figure 4). This hypothesis is further supported by figure 3, which shows that virtually all experimental threads in our data set, which are likely to transport sediment, are wider, or

10  as wide as, the threshold channel. This observation suggests that the theory of Glover and Florey corresponds to the narrowest possible channel, which forms in the absence of sediment transport (Henderson, 1963; Parker, 1978b). We hypothesize that, as the latter increases, the channel's width departs from this lower boundary. Unfortunately, Stebbings did not measure sediment discharge in his channels, and we cannot quantify the dependency of the channel's width with respect to sediment discharge.

     What Stebbings did measure, though, is the channel's width at the onset of braiding, just upstream of the first bar (figure 4).

15  We refer to this value as the "limit-channel width", implying it corresponds to the largest possible width of a stable channel. Once detrended according to eq. (2), the limit-channel width $W_{*,l}$ shows no remaining correlation with discharge (figure 5), indicating that it is proportional to the width of the threshold channel. The proportionality factor is about $\langle W_{*,l} \rangle = 1.7 \pm 0.2$, thus significantly larger than one. The detrended limit-channel width is narrowly distributed around its own average, much like the threshold-channel width (figure 5). The two average values are clearly distinct, to the 95% level of confidence. In short, the channel destabilizes into a braid when it gets about 1.7 times as large as the threshold channel.

     Based on this observation, we propose the following scenario for the transient in Stebbings' experiments. As its upstream end incises the sediment layer, the river loads itself with sediment. The continuous increase of bedload transport along its





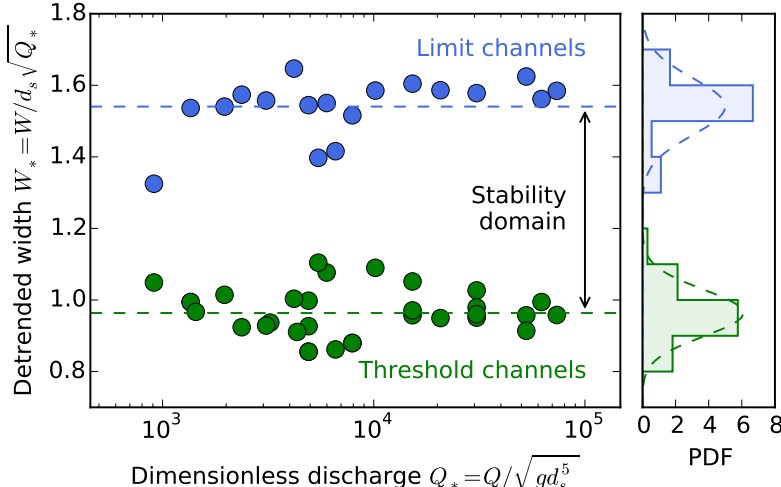

**Figure 5.** Detrended channel width in Stebbings' laboratory experiments (Stebbings, 1963). Green: threshold channels (no sediment transport); blue: active channels about to split . Left: detrended width $W_*$ as a function of dimensionless discharge; right: normed histograms of the same data. Dashed lines indicate fitted Gaussian distributions.

course causes it to widen, until it reaches the limit-channel width. At this point, bars develop and quickly split the river into multiple channels. Generalizing this interpretation, we suggest that a river can only accommodate so much sediment transport before it breaks into a braid. This fragility would confine single-thread channels to a precarious domain in the parameter space, thus explaining their rarity in laboratory experiments.

To our knowledge, only Ikeda et al. (1988) produced active and stable single-thread channels in a laboratory experiment. To do so, they first carved an initially straight channel in non-cohesive sediment. To prevent the formation of bars, and the lateral migration of the channel, Ikeda et al. cut the channel in half with a vertical wall aligned with the channel's axis. Water and sediment are then injected at constant rate. Eventually, this experiment generates a stable half channel with a flat lower section were sediment are continuously transported. (Hereafter, we use twice the width of the half channel, for comparison with other experiments.)

It is unclear whether the channels of Ikeda et al. have fully reached steady state, with as much sediment exiting the experiment as is injected into it. Nonetheless, the actual sediment discharge appears to be low enough to allow for stable channels, which we may treat as a collection of single-thread active channels. Their detrended width is distributed narrowly around a mean value of $\langle W_{*,s} \rangle = 1.16 \pm 0.16$ (figure 6). As expected, this value falls within the stability domain based on Stebbings' experiments, to the 95% level of confidence (figures 5 and 6). Based on the report by Ikeda et al. only, we cannot be certain that no stable channel could survive outside the stability domain. Neither can we evaluate the influence of the central wall on the channel's stability. However, these observations are clearly consistent with our interpretation of Stebbings' experiment.





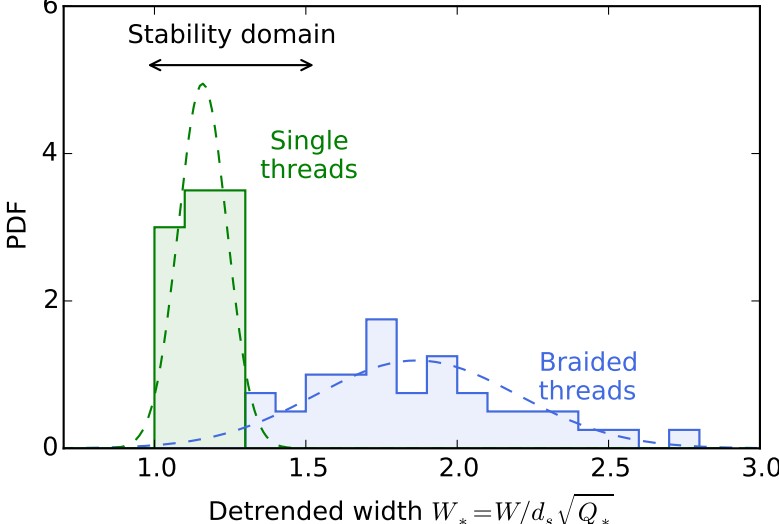

**Figure 6.** Normed histograms of the detrended width of laboratory channels. Green: single-thread channels (Ikeda et al., 1988), blue: threads of braided rivers (Ashmore, 2013).

Stebbings' observations suggest that single-thread channels destabilized by sediment transport become braids. The mechanism by which this metamorphosis occurs is still a matter of debate, although the bar instability has been repeatedly pointed at (Parker, 1976; Devauchelle et al., 2010b, a). What is likely, though, is that once the river has turned into a braid, each of its channels transports only a fraction of the total sediment discharge. It is therefore reasonable to treat it as an active channel itself, and compare its width to the threshold theory. This method was applied with some success to natural braided rivers, and in section 2 (Gaurav et al., 2015; Métivier et al., 2016).

In his review on braided rivers, Ashmore (2013) reports on laboratory experiments he performed in the 1980's. What makes his experiments unique is that he measured the size and the discharge of the individual threads that compose his braided rivers. Translating his measurements in terms of the detrended width $W_{*,b}$, we find that its distribution spreads around an average of $\langle W_{*,b} \rangle = 1.87 \pm 0.68$, close to the upper bound of the stability domain (figure 6). One way to interpret this observation, although speculative at this point, is to consider the upper bound of the stability domain as an attractor for the threads' dynamics. Accordingly, we conjecture that the threads of a braided river, constantly destabilized by an excessive sediment discharge, split into smaller channels. These channels, when numerous enough, are likely to meet one another and recombine their sediment load. This process could repeat itself until reaching the dynamical equilibrium which characterizes a braided river (Métivier and Meunier, 2003; Reitz et al., 2014). The thread population resulting from this equilibrium would include stable channels, which detrended width lies in the stability domain, and splitting channels, which we expect to be wider than the limit channel. The broad distribution of $W_{*,b}$ in Ashmore's experiment is consistent with this interpretation (figure 6), as are the center bars often found in the threads of natural braided rivers (Gaurav et al., 2015; Métivier et al., 2016).




## 4 Conclusion

More than a hundred years of laboratory investigations have improved our understanding of how river select their own morphology. Here, we have revisited some of these experiments to place them in the perspective of the threshold theory introduced by Glover and Florey (1951) and Henderson (1963). Although these experiments were designed to investigate a variety of phenomena, the channels they produced all conform to Lacey's law, exactly like natural rivers. This indicates that laboratory flumes and natural rivers are indeed controlled by the same primary mechanisms, in accordance with Jaggar's views. We take it as encouragement for experimental geomorphology.

Most laboratory channels are larger than predicted by the threshold theory. Based on the experiment of Stebbings (1963), we propose that, for the most part, sediment transport induces this departure from the threshold channel. According to this interpretation of Stebbings' observations, the channel widens to accommodate more bedload, until it reaches a width of about 1.7 times that of the threshold channel, at which point it destabilizes into a braided river. The writing of Stebbings' paper suggests that, had he been aware of the work of Glover and Florey (1951), he would have drawn similar conclusions from his experiment. To our knowledge, the influence of the sediment discharge on the width of a channel has never been measured directly (Stebbings did not measure the sediment discharge). The laboratory would certainly be a convenient locale to do so.

Mentions of active single-thread channels are scarce in the literature on laboratory rivers, although some authors succeeded in maintaining such channels by various means, such as riparian vegetation or cohesive sediment. More often, though, laboratory flumes generate braided rivers. Again, we suspect sediment discharge is the real culprit for this familiar destabilization. Accordingly, it should be possible to produce active and stable single-thread channels simply by lowering enough the sediment input. If this method works, not only will we be able to quantify the influence of sediment transport on a channel's width, but it will also gain us a laboratory rat for single-thread rivers. We believe it would shed light on the dynamics of such rivers, including meandering.

*Acknowledgements.* We gratefully acknowledge P. Ashmore for sharing his database on experimental braided threads. O.D. was funded by the *Émergence(s)* program of the *Mairie de Paris*, France. This is IPGP contribution #3758.




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



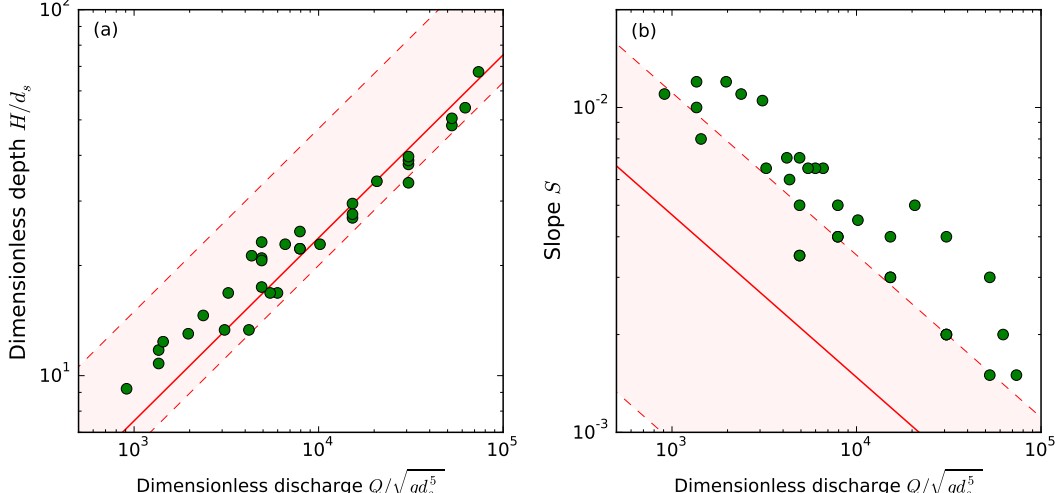

**Figure 7.** Regime relationship for the depth **(a)** and slope **(b)** measured in the experiment of Stebbings (1963). Solid red line corresponds to the threshold theory (eq. (1) with $\theta_t = 0.05$ and $C_f = 0.1$). Shaded area and dashed lines indicate uncertainty about the parameters in Stebbings' experiment.

Tal, M. and Paola, C.: Dynamic single-thread channels maintained by the interaction of flow and vegetation, Geology, 35, 347–350, 2007.

Tal, M. and Paola, C.: Effects of vegetation on channel morphodynamics: results and insights from laboratory experiments, Earth Surf. Proc. Landf., 35, 1014–1028, 2010.

Warburton, J.: A brief review of hydraulic modelling of braided gravel bed rivers, J. Hydrol. New Zealand, 35, 157–174, 1996.

5 **Appendix A:  Threshold theory for depth and slope**

In addition to the width, the threshold theory provides a estimate for the depth and the slope of channel at threshold (Glover and Florey, 1951; Henderson, 1963; Devauchelle et al., 2011b; Seizilles, 2013):

$$\frac{H}{d_s} = \left[ \frac{\sqrt{\mu}}{\pi} \left( \frac{\theta_t(\rho_s - \rho)}{\rho} \right)^{-1/4} \sqrt{\frac{3\sqrt{2}C_f}{\mathcal{K}[1/2]}} \right] Q_*^{1/2}, \tag{A1}$$

and

10  $$S = \left[ \sqrt{\mu} \left( \frac{\theta_t(\rho_s - \rho)}{\rho} \right)^{5/4} \sqrt{\frac{\mathcal{K}[1/2]\, 2^{3/2}}{3C_f}} \right] Q_*^{-1/2}. \tag{A2}$$

We now compare these regime equations to Stebbings' experimental channels (figure 7a). The depth of the channels accords with equation (A1), although with slightly more scatter around the prediction than for the width (figure 3). Measurement uncertainty probably explain this dispersion, since the depth of a channel is less accessible than its width.



The downstream slope of Stebbings' channel appears more dispersed than the width (figure 7b). The corresponding data points nonetheless follow a clear power law, compatible with the inverse square root predicted by eq. (A2). The prefactor of this relationship, however, falls around the upper bound of the uncertainty range. We do not know the origin of this offset, for which we can only propose speculative explanations. First, as the slope of experimental channels is notoriously difficult to measure, a

5 systematic error cannot be ruled out (Stebbings provides no indication about the accuracy of his slope measurements). Second, as readily seen by comparing equations (1) and (A2), the slope of a threshold channel is sensitive to the value of the threshold Shields parameter. A value twice as large would account for Stebbings' slope measurements, without impacting significantly the width and depth of the threshold channel. Finally, to our knowledge, the regime equations of a channel at threshold have always been established using the shallow-water approximation. In real channels, the flow transfers momentum across the

10 stream (Parker, 1978b). Taking this transfer into account could correct he threshold theory, without altering much the scalings it predicts