# Peer review of "Laboratory rivers: Lacey's law, threshold theory and channel stability"

_Earth Surface Dynamics, 2016_

## Referee Comment (RC1) · D. Jerolmack (Referee) · 29 Sep 2016

Reviewer: Douglas Jerolmack

This paper synthesizes 100+ years of research on laboratory river channel geometry data to assess the success of threshold channel theory for explaining the relation between river discharge and width. It also seeks to better our understanding of why braided rivers are so common in the lab, and whether there is a phase space in which single-thread channels that transport sediment can exist WITHOUT the assistance of bank cohesion/vegetation. The main conclusions from this assessment of experiments are that: (1) threshold channel theory predicts very well the width of laboratory channels with no or little sediment transport; (2) threshold channel theory predicts the scaling relation between width and discharge well for lab rivers with significant transport,

but these rivers are offset from the threshold prediction; and (3) increasing sediment transport rate leads to river widening and a departure from the threshold prediction until a critical width - likely associated with bar formation - at which the river begins to braid. The first two points have already been demonstrated by some of these same researchers for field rivers; as they point out, it is quite nice that lab rivers behave identically in this respect to field rivers. The third point is the most novel and also speculative finding of this paper, but it is backed up by a fascinating re-analysis of experimental data. In particular, the authors look at the width of channels just before the point of bar formation in some classic transient channel adjustment experiments from the 1960s, and show that there appears to be a well defined upper limit on channel width.

The paper is clear, concise, and well written. The analysis is straightforward but fascinating. The findings about the upper limits of channel width stability raise more questions than they answer, but this also tells us precisely where to go with future experiments. Indeed, the authors usefully suggest some specific future experiments that could shed light on these questions. I don't see any need for major revisions. However, these findings touch on other questions and I found myself wondering if and how their interpretation may agree or disagree with other studies. It might be useful for the authors to broaden out their discussion to include some of these. First, the conditions for bar iniation and geometry are well known and theories and laboratory experiments are in pretty good agreement. Is the finding in this paper of channel width = 1.7 times threshold for braiding related the classically determined bar initiation? Related, another classic Parker paper on the braiding/meandering transition proposes a stability criterion related to water discharge, width, depth and slope that has been validated with field data; this is related to the onset of bar formation. Does the authors' result here square with the Parker stability criterion? More broadly, this paper steers clear of the question of how far rivers can get from threshold - it does not consider the Shields stress of rivers in the analysis, and considers width-discharge scaling (and depth and slope in the appendix). Do the findings in this paper bear at all on the Shields stress of rivers? The paper cited for field alluvial river data, Li et al. (2015), proposes a continuously varying
Shields stress and indicates that there is no real attractor of (near-)threshold dynamics. On the other hand, we (Phillips and Jerolmack, 2016) assessed coarse-grained rivers - including some of the same data - and found that when the dependence of critical Shields stress on channel slope is taken into account, there is strong evidence that channels organize to a Shields stress slightly in excess of critical. It is indeed fascinating that the threshold channel theory predicts the proper scaling of channel geometry with discharge, even when the magnitude is incorrect (as shown by these authors in their previous 2015 paper from field data, and also in the appendix of this paper where we find that depth and especially slope are offset from the theoretical line). Do the authors have any comment on what we can say about the state of Shields stress of rivers? In other words, is the existence of a certain transport state (Shields stress or Rouse number) a cause or a consequence of channel organization? I know the data alone can't answer this question, but these authors already flirt with some of the deep paradoxes of channel geometry, and are in a better position than most to speculate on this too.

Minor comments follow below. p. 6, line 20: "lead" should be changed to "led".

Table 1: Could you provide parameters that would allow computation of hydraulic variables? e.g., Discharge, width, depth, slope?

p. 8, line 12: "anymore" should be "any more" line 15: the word "influence" is mispelled.

Figure 5 caption: "normed" should be "normalized".

p. 10, about the rarity of single channels in the lab and possibly very small phase space. Could the authors comment on whether there is something about field scale that makes this phase space "less fragile"? Or in the field is it cohesion/vegetation that foreces this?

p. 10, line 9: authors point out that Ikeda is only experiment they know of that makes stable single-thread rivers in the lab. They should make clear that they mean without

adding effects of cohesion/vegetation.

p. 10: "...with a lower flat section were...". The "were" should be "where".

**ESurfD**

---

## Referee Comment (RC2) · D.R Parsons (Referee) · 29 Nov 2016

Overall comments: The paper draws together a range of previous studies on laboratory channels to explore the evolution o channel geometry data in order to assess examine the ability of threshold channel theory in explaining relationships between discharge and width. The paper is well written and presented and examines and explores why many laboratory channels evolve to braided rivers when physical or biological cohesive effects are not included. The work concludes that threshold channel theory can predict laboratory channel width when there is low sediment flux rates but that as flux rates increase a critical rate is reached when bar formation processes begin and the theory begins to break down at the onset of braiding. This is the main outcome from the paper and is an interesting and generally novel set of ideas, albeit a little speculative at times! The discussion needs some attention in two main parts. 1) I think the paper would ben-

**ESurfD**
* * *
Interactive
comment

efit from including a section that examines the interrelationships between width, depth, slope and shear stress. . .. as ultimately these have been shown to control alternate bar formation. . . inclusion of the work of Mosselman and Crossato, recent work by van der Lageweg would be valuable as well as the classic work of Parker on thresholds between braiding and meandering. Either this older theory is unnecessarily complex or the analysis herein too simple. . .a discussion of this would be welcome addition. 2) I would also welcome an additional element in the discussion on bank and substrate cohesion and its influence on geometry and sediment transport processes, including bedform and bar form development – with impacts on sediment transport rate and shear stress. Do more cohesive sediments inhibit bar formation and maintain single thread channels as some have argued. . . is this bank stability or bedform suppression (e.g. Schindler et al. 2016). Artificiality holding shear and depth higher? There is a wealth of additional work examining these elements that should be included and would result in a much more widely used paper. I have a few minor comments: i) Rephrase sentence in abstract or break up sentence: "Using this finding to reinterpret experiments by Stebbings (1963), we suggest that sediment transport widens the channel until it reaches a limit width, beyond which it destabilizes into a braided river." ii) page 6, 20: "led". iii) page 8, 12: "anymore" iv) Page 8, 15: "influence"

Daniel R. Parsons

---

## Author Comment (AC1) · 27 Feb 2017

**Answers to Prof. Jerolmack**

February 27, 2017

This paper synthesizes 100+ years of research on laboratory river channel geome- try data to assess the success of threshold channel theory for explaining the relation between river discharge and width. It also seeks to better our understanding of why braided rivers are so common in the lab, and whether there is a phase space in which single-thread channels that transport sediment can exist WITHOUT the assistance of bank cohesion/vegetation. The main conclusions from this assessment of experiments are that: (1) threshold channel theory predicts very well the width of laboratory chan- nels with no or little sediment transport; (2) threshold channel theory predicts the scal- ing relation between width and discharge well for lab rivers with significant transport, but these rivers are offset from the threshold prediction; and (3) increasing sediment transport rate leads to river widening and a departure from the threshold prediction until a critical width - likely associated with bar formation - at which the river begins to braid. The first two points have already been demonstrated by some of these same researchers for field rivers; as they point out, it is quite nice that lab rivers behave identi- cally in this respect to field rivers. The third point is the most novel and also speculative finding of this paper, but it is backed up by a fascinating re-analysis of experimental data. In particular, the authors look at the width of channels just before the point of bar formation in some classic transient channel adjustment experiments from the 1960s, and show that there appears to be a well defined upper limit on channel width. The paper is clear, concise, and well written. The analysis is straightforward but fascinating. The findings about the upper limits of channel width stability raise more questions than they answer, but this also tells us precisely where to go with future experiments. Indeed, the authors usefully suggest some specific future experiments that could shed light on these questions.

I don't see any need for major revisions. However, these findings touch on other questions and I found myself wondering if and how their interpretation may agree or disagree with other studies. It might be useful for the authors to broaden out their discussion to include some of these.

Detailed answers to your comments and questions are given below. When two questions are related we have grouped them together and provided an overall answer.

First, the conditions for bar iniation and geometry are well known and theories and laboratory experiments are in pretty good agreement. Is the finding in this paper of channel width = 1.7 times threshold for braiding related the classically determined bar initiation ? Related, another classic Parker paper on the braiding/meandering transition proposes a stability crite- rion related to water discharge, width, depth and slope that has been validated with field data; this is related to the onset of bar formation. Does the authors' result here square with the Parker stability criterion ?

Parker (1976) studied the formation of bars and its consequences on channel geometry. He proposed that the limit between single-thread and multiple-thread channels occurs when the ratio (H/W) of a stream is equal to the ratio of its slope to the Froude number of the flow (S/Fr). Following your suggestion, we compared our limit-channels with this criterion and found that the limit-channels of Stebbings are on the verge of braiding according to Parker's prediction. Furthermore we also find that the threshold channels and the experimental channels of Ikeda et al (1988) and Ashmore (2013) accord with the threshold proposed by Parker. Thus our empirical finding, based on these experiments, is not in contradiction with Parker's analysis. We added a paragraph to explain this in detail in the text and a supplement in appendix with a new figure that shows the comparison.

More broadly, this paper steers clear of the question of how far rivers can get from threshold - it does not consider the Shields stress of rivers in the analysis, and considers width-discharge scaling (and depth and slope in the ap- pendix). Do the findings in this paper bear at all on the Shields stress of rivers?

The paper cited for field alluvial river data, Li et al. (2015), proposes a continuously varying Shields stress and indicates that there is no real attractor of (near-)threshold dynamics. On the other hand, we

(Phillips and Jerolmack, 2016) assessed coarse-grained rivers - including some of the same data - and found that when the dependence of critical Shields stress on channel slope is taken into account, there is strong evidence that channels organize to a Shields stress slightly in excess of critical. It is indeed fascinat- ing that the threshold channel theory predicts the proper scaling of channel geometry with discharge, even when the magnitude is incorrect (as shown by these authors in their previous 2015 paper from field data, and also in the appendix of this paper where we find that depth and especially slope are offset from the theoretical line). Do the authors have any comment on what we can say about the state of Shields stress of rivers? In other words, is the existence of a certain transport state (Shields stress or Rouse number) a cause or a consequence of channel organization? I know the data alone can't answer this question, but these authors already flirt with some of the deep paradoxes of channel geometry, and are in a better position than most to speculate on this too.

In a threshold river the channel shape adjusts so that the shields parameter is everywhere constant and equal to $\theta_t$. When the shear stress is above threshold, theories have been proposed to explain the geometry of the section but the available experiments do not allow us to test these theories. Specifically, we lack experiments with shear stress measurements. They would be needed to provide a sound answer to your question. Of course we agree that this is an important questions and it is the subject of an ongoing experimental and theoretical work in our group.

Minor comments follow below.

p. 6, line 20: "lead" should be changed to "led". Done

Table 1: Could you provide parameters that would allow computation of hydraulic variables? e.g., Discharge, width, depth, slope?

p. 8, line 12: "anymore" should be "any more" line 15: the word "influence" is mispelled. Done

Figure 5 caption: "normed" should be "normalized". Done

p. 10, about the rarity of single channels in the lab and possibly very small phase space. Could the authors comment on whether there is something about field scale that makes this phase space "less fragile"? Or in the field is it cohesion/vegetation that forces this?

p. 10, line 9: authors point out that Ikeda is only experiment they know of that makes stable single-thread rivers in the lab. They should make clear that they mean without adding effects of cohe- sion/vegetation.

We changed the sentence.

p. 10: "...with a lower flat section were...". The "were" should be "where". Done

---

## Author Response (AR1)

François Métivier
IPGP 1 rue Jussieu
75005 Paris, France

February 27, 2017

Andreas Lang, Editor

Dear Andreas

You will find enclosed

- our detailed answers to the reviews of Prof. Jerolmack and Prof. Parsons,
- a difference file displaying the changes we made to the manuscript in response to the comments

In their comments D. Jerolmack and D. Parsons raised the following three main points.

1. Both reviewers wonder how our analysis and prediction compares with theoretical analyses of the meandering to braiding transition.
2. D. Parsons asked us to include a discussion on the effect of substrate cohesion and its potential influence on our analysis.
3. D. Jerolmack raised the problem of the relationship between the state of stress and its relationship to the geometry of the channel when a river is above threshold conditions.

The first point led us to compare our analysis to the fundamental work of Parker in the seventies. Using the same dataset, we were able to show that our re-analysis of existing experiments is not in contradiction with the theory proposed by Parker, and that this brings confidence in our work. The second point led us to discuss the reasons that may explain the dispersion we observe around the trend predicted by the threshold theory. The last point led us to acknowledge in our answer that, given the present state of experimental geomorphology, more experiments are needed to address this fundamental question.

Overall we found both reviews to be very positive and helpful. The few points raised were important ones that helped us strengthen the content of the manuscript.

We hope you will now find this contribution satisfying, and, again, let me, on behalf of my two co-authors, apologize for the time it took us to come up with this corrected manuscript.

Yours faithfully

François Métivier

**Answers to Prof. Jerolmack**

February 27, 2017

This paper synthesizes 100+ years of research on laboratory river channel geome- try data to assess the success of threshold channel theory for explaining the relation between river discharge and width. It also seeks to better our understanding of why braided rivers are so common in the lab, and whether there is a phase space in which single-thread channels that transport sediment can exist WITHOUT the assistance of bank cohesion/vegetation. The main conclusions from this assessment of experiments are that: (1) threshold channel theory predicts very well the width of laboratory chan- nels with no or little sediment transport; (2) threshold channel theory predicts the scal- ing relation between width and discharge well for lab rivers with significant transport, but these rivers are offset from the threshold prediction; and (3) increasing sediment transport rate leads to river widening and a departure from the threshold prediction until a critical width - likely associated with bar formation - at which the river begins to braid. The first two points have already been demonstrated by some of these same researchers for field rivers; as they point out, it is quite nice that lab rivers behave identi- cally in this respect to field rivers. The third point is the most novel and also speculative finding of this paper, but it is backed up by a fascinating re-analysis of experimental data. In particular, the authors look at the width of channels just before the point of bar formation in some classic transient channel adjustment experiments from the 1960s, and show that there appears to be a well defined upper limit on channel width. The paper is clear, concise, and well written. The analysis is straightforward but fascinating. The findings about the upper limits of channel width stability raise more questions than they answer, but this also tells us precisely where to go with future experiments. Indeed, the authors usefully suggest some specific future experiments that could shed light on these questions.

I don't see any need for major revisions. However, these findings touch on other questions and I found myself wondering if and how their interpretation may agree or disagree with other studies. It might be useful for the authors to broaden out their discussion to include some of these.

Detailed answers to your comments and questions are given below. When two questions are related we have grouped them together and provided an overall answer.

First, the conditions for bar iniation and geometry are well known and theories and laboratory experiments are in pretty good agreement. Is the finding in this paper of channel width = 1.7 times threshold for braiding related the classically determined bar initiation ? Related, another classic Parker paper on the braiding/meandering transition proposes a stability crite- rion related to water discharge, width, depth and slope that has been validated with field data; this is related to the onset of bar formation. Does the authors' result here square with the Parker stability criterion ?

Parker (1976) studied the formation of bars and its consequences on channel geometry. He proposed that the limit between single-thread and multiple-thread channels occurs when the ratio (H/W) of a stream is equal to the ratio of its slope to the Froude number of the flow (S/Fr). Following your suggestion, we compared our limit-channels with this criterion and found that the limit-channels of Stebbings are on the verge of braiding according to Parker's prediction. Furthermore we also find that the threshold channels and the experimental channels of Ikeda et al (1988) and Ashmore (2013) accord with the threshold proposed by Parker. Thus our empirical finding, based on these experiments, is not in contradiction with Parker's analysis. We added a paragraph to explain this in detail in the text and a supplement in appendix with a new figure that shows the comparison.

More broadly, this paper steers clear of the question of how far rivers can get from threshold - it does not consider the Shields stress of rivers in the analysis, and considers width-discharge scaling (and depth and slope in the ap- pendix). Do the findings in this paper bear at all on the Shields stress of rivers?

The paper cited for field alluvial river data, Li et al. (2015), proposes a continuously varying Shields stress and indicates that there is no real attractor of (near-)threshold dynamics. On the other hand, we

**Answers to Prof. Parsons**

February 27, 2017

Overall comments: The paper draws together a range of previous studies on laboratory channels to explore the evolution o channel geometry data in order to assess examine the ability of threshold channel theory in explaining relationships between discharge and width. The paper is well written and presented and examines and explores why many laboratory channels evolve to braided rivers when physical or biological cohesive effects are not included. The work concludes that threshold channel theory can predict laboratory channel width when there is low sediment flux rates but that as flux rates increase a critical rate is reached when bar formation processes begin and the theory begins to break down at the onset of braiding. This is the main outcome from the paper and is an interesting and generally novel set of ideas, albeit a little speculative at times!

The discussion needs some attention in two main parts.

1) I think the paper would benefit from including a section that examines the interrelationships between width, depth, slope and shear stress. . .. as ultimately these have been shown to control alternate bar formation. . . inclusion of the work of Mosselman and Crossato, recent work by van der Lageweg would be valuable as well as the classic work of Parker on thresholds between braiding and meandering. Either this older theory is unnecessarily complex or the analysis herein too simple. . .a discussion of this would be welcome addition.

Following your suggestion we have added a discussion on that matter and a comparison to the stability analysis of Parker (1976), who was the first to examine the interrelationship between width, depth, slope and Froude number. Parker proposed that the limit between meandering and braiding channels occurs when the ratio (H/W) of a stream is of the same order of magnitude as the ratio of Slope of the channel to the Froude number of the flow (S/Fr). We compared our prediction with Parker's prediction using the same dataset and found that they are in good agreement. Comparison with more recent analyses, such as the one you mention, is difficult given the available data but the comparison with Parker's criterion undoubtedly shows that our empirical analysis is not in opposition with these theoretical analyses. We added a paragraph in the discussion and an appendix with a figure showing this comparison.

2) I would also welcome an additional element in the discussion on bank and substrate cohesion and its influence on geometry and sediment transport processes, including bedform and bar form development – with impacts on sediment transport rate and shear stress. Do more cohesive sediments inhibit bar formation and maintain single thread channels as some have argued. . . is this bank stability or bedform suppression (e.g. Schindler et al. 2016). Artificiality holding shear and depth higher? There is a wealth of additional work examining these elements that should be included and would result in a much more widely used paper.

Vegetation and bank cohesion indeed influence the geometry of the channel. Our analysis reveals that, at leading order, the geometry of experimental rivers accords with the threshold theory. Deviations of factors up to two can be seen from this scaling on figures 2 and 3. These deviations can be caused by the additional phenomena you mention. Tal and Paola (2007,2010), for example, have shown that vegetation could significantly lower the aspect ratio of a thread. The same has been done for cohesive sediments by Peakall et al. (2007) and Dijk et al. (2012). We have edited the manuscript to make this point clearer.

I have a few minor comments:

[revised manuscript text omitted]